# Comparison of Different Techniques to Calculate Properties of Atmospheric Turbulence from Low-Resolution Data

**Marta Wacławczyk** [1,*], **Amoussou S. Gozingan** [2], **Jackson Nzotungishaka** [2], **Moein Mohammadi** [1] **and Szymon P. Malinowski** [1]

[1] Institute of Geophysics, Faculty of Physics, University of Warsaw, Pasteura 5, 02-093 Warsaw, Poland; moein.mohammadi@fuw.edu.pl (M.M.); Szymon.Malinowski@fuw.edu.pl (S.P.M.)
[2] African Institute for Mathematical Sciences, Summerhill Estates, East Legon Hills, Santoe, Accra GA184, Ghana; amoussou@aims.edu.gh (A.S.G.); jackson@aims.edu.gh (J.N.)
[*] Correspondence: marta.waclawczyk@fuw.edu.pl

**Abstract:** In this work we study different techniques to estimate basic properties of turbulence, that is its characteristic velocity and length scale from low-resolution data. The methods are based on statistics of the signals like the velocity spectra, second-order structure function, number of signal's zero-crossings and the variance of velocity derivative. First, in depth analysis of estimates from artificial velocity time series is performed. Errors due to finite averaging window, finite cut-off frequencies and different fitting ranges are discussed. Next, real atmospheric measurement data are studied. It is demonstrated that differences between results of the methods can indicate deviations from the Kolmogorov's theory or the presence of external intermittency, that is the existence of alternating laminar/turbulent flow patches.

**Keywords:** turbulence; clouds; intermittency; turbulence kinetic energy; turbulence kinetic energy dissipation rate

## 1. Introduction

Turbulence is the most common regime of fluid flow. It is characterised by its high rotationality, enhanced mixing of mass, momentum and energy and, most importantly, a myriad of motions, called eddies, of different length and time scales, strongly interacting with each other [1]. Turbulence is not only a flow regime, but it is also a driving factor of transport and mixing and, hence, influences e.g., the boundary layer formation and affects air pollution episodes [2]. While in many flows turbulence and its effects can be investigated in a systematic way by controlled experiments, this is not the case for atmospheric turbulence. Effects of turbulent transport and mixing in scales smaller than the gridbox of numerical simulations are still major limitations in weather and climate simulations [3]. Despite many airborne measurements and research campaigns our understanding of turbulence in free atmosphere is still far from sufficient. Part of the problem is the limited amount of measurement data, another part is measurement errors, and last but not least element is inadequate or unsatisfactory data analysis.

One of the reasons that atmospheric turbulence is still a major problem is that it is in general neither stationary nor isotropic [4–6]. It can be generated by wind shear and/or buoyancy forces acting locally (e.g., due to diabatic heating), can be transported by large scale flows and circulations, and affected by stratification [7] as well as external intermittency (i.e., presence of alternating laminar and turbulent regions). This fact affects estimates of turbulence properties from measurements as well as deteriorates predictions of turbulence models [8].

The simplest way to characterize turbulence, widely used in turbulence-closure models, is to define characteristic velocity $U$ and length $L$ (or time $T$) scales of turbulent eddies. Two quantities necessary to estimate them are the turbulence kinetic energy $\mathcal{K}$ and the turbulence kinetic energy dissipation rate (EDR) $\epsilon$. While $\mathcal{K}$ is determined mainly by the large, energy-containing eddies, $\epsilon$ is defined as the rate of conversion of the turbulence kinetic energy into thermal energy, which takes place at small scales. In atmospheric turbulence these scales are of order of milli- or centimeter, while the resolution of signals available from airborne measurements is typically smaller by 3–4 orders of magnitude.

In practice, $\epsilon$ can be estimated from measured signals based on the assumption of local isotropy and the Kolmogorov's hypotheses [9]. The second hypothesis states that in a certain range of wavenumbers, called the inertial range, statistics of turbulence, such as the wavenumber spectrum functions, have a universal scale-invariant form and are determined only by the rate of dissipation $\epsilon$. This allows to estimate EDR from the measured inertial-range part of the power spectrum of velocity fluctuations. However, due to limitations of sensors, measurement errors and rapid changes of atmospheric conditions along the flight track, $\epsilon$ estimates from airborne data are far from being standard. It can be expected that different approaches used to estimate EDR will not respond identically to various types of error. The first objective of the present paper is to test and compare different methods to determine $\epsilon$ from experimental data. Using different estimates will help to reduce errors of the calculated $\epsilon$. Moreover, the aim of this work is to establish a list of guidelines to specify which method(s) should be used under certain circumstances (e.g., in the case of low-frequency measurements or short fitting ranges). In this study, we will analyse the standard methods for EDR estimate from power spectra, second-order structure functions, as well as the recently proposed techniques based on the number of signal's zero-crossings and iterative methods [10,11]. The iterative methods are based on the second as well as the first Kolmogorov's hypothesis. The first hypothesis states that at sufficiently high Reynolds numbers statistics of small-scale motions have a universal form, uniquely determined by $\epsilon$ and the molecular viscosity of the fluid $\nu$. In the new method the missing, unmeasured part of the spectrum is reconstructed down to the smallest dissipative eddies using the signal from the resolved measurements.

All the techniques are tested first on the artificial velocity time series with the use of different averaging windows, different cut-off frequencies and fitting ranges. Next, the same tests are repeated on measurement data obtained during a horizontal flight through the stratocumulus cloud within the POST (Physics of the Stratocumulus Cloud-Top) campaign [12–15].

Apart from the reduction of the statistical error, comparison between obtained results may deliver new information on the properties of the investigated atmospheric turbulence, especially its deviations from the Kolmogorov's scaling and/or the presence of external intermittency. It is shown that in the former case EDR estimated from the structure function and the iterative method are under or over-predicted in comparison to the remaining results. On the other hand, under-prediction of the zero-crossings indicate the presence of external intermittency or strong large-scale motions. This fact is prospective for further applications, e.g., in the analysis of stratocumulus cloud turbulence, as it could allow to detect anisotropic zones and the existence of non-turbulent flow patches due to entrainment of clear-air or local relaminarisation [16]. We can expect, EDR estimates in such regions are deteriorated [11], as the Kolmogorov's assumptions are not satisfied.

The present paper is structured as follows, the applied methods are discussed in Section 2. Error analysis of EDR estimates is described in Section 3 and results calculated from artificial and real signals are presented in Sections 4 and 5, respectively. Detection of deviations from the Kolmogorov's scaling and the presence of external intermittency is investigated in Section 6. This is followed by the discussion of the obtained results in Section 7 and the Conclusions.

## 2. Description of Methods

### 2.1. Characteristic Scales of Turbulence

The turbulence kinetic energy $\mathcal{K}$ and the turbulence kinetic energy dissipation rate are defined, respectively, as [17]

$$\mathcal{K} = \frac{1}{2}\langle u_i u_i \rangle, \quad \epsilon = 2\nu \langle s_{ij} s_{ij} \rangle \tag{1}$$

where $s_{ij} = 1/2(\partial u_i/\partial x_j + \partial u_j/\partial x_i)$, $\langle \cdot \rangle$ is the ensemble average and $u_i = U_i - \langle u_i \rangle$ denotes the $i$-th component of fluctuating velocity. With these two quantities characteristic velocity and length scales can be estimated. Dimensional analysis provides [18]

$$\mathrm{U} = \sqrt{\mathcal{K}}, \quad \mathcal{L} = C_\epsilon \frac{\mathcal{K}^{3/2}}{\epsilon}, \tag{2}$$

where $C_\epsilon$ was assumed constant till recent studies of J. C. Vassilicos [19] who generalized the Taylor's formula (2). Equation (2) can be considered as the rough estimate of turbulence properties and are a basis of common turbulence models, which neglect the multiscale character of turbulence. Apart from this, the dissipation $\epsilon$ and the kinematic molecular viscosity $\nu$ allow to estimate the so-called Kolmogorov scales, i.e., the characteristic length, velocity and the time scales of the smallest vortex structures

$$\eta = (\nu^3/\epsilon)^{1/4}, \quad u_\eta = (\nu\epsilon)^{1/4}, \quad \tau = (\nu/\epsilon)^{1/2}. \tag{3}$$

These small scales are of utmost importance for the analysis of physical processes in clouds, as they influence e.g., the collision frequency of water droplets and consequently the time needed for precipitation to occur [20–22].

Since direct airborne measurements of the dissipation rate are not possible, the so-called indirect methods are used. The basis for all the methods is the Kolmogorov's local isotropy hypothesis [9], which relies on the idea that anisotropy present at large scales is lost as the energy is transferred to smaller eddies in the cascade process. As a consequence, two-point turbulence statistics at small scales do not depend on the direction of the vector joining the two points, but only on its magnitude. With this, it is possible to estimate the dissipation rate from 1D intersections of the 3D velocity field.

The Kolmogorov's first similarity hypothesis divides the wavenumber space into the energy-containing and the universal equilibrium range. Under the assumption of local isotropy it was inferred that statistics of turbulent motions in the latter range have a universal form, uniquely determined by $\nu$ and $\epsilon$. With this, based on dimensional arguments the energy spectrum in the equilibrium range can be described by the formula

$$E(\kappa) = \epsilon^{2/3}\kappa^{-5/3}f(\eta\kappa), \tag{4}$$

where $\kappa$ is the wavenumber. The second similarity hypothesis divides the universal equilibrium range into two sub-ranges: the dissipation and the inertial subrange. In the latter the statistics are uniquely determined by $\epsilon$, independent of $\nu$, which suggests that $f(\eta\kappa)$ in Equation (4) becomes a constant and

$$E(\kappa) = C\epsilon^{2/3}\kappa^{-5/3}. \tag{5}$$

Hence, the idea of Kolmogorov enables to estimate $\epsilon$ and thus, the characteristic scales of small eddies based on statistics of inertial-range motions. With this, the EDR of atmospheric turbulence can be estimated from in-situ measurements even though the research aircrafts are often not equipped to measure wind fluctuations with resolutions better than a few tens of meters. This length is much above the typical millimiter-order Kolmogorov length.

### 2.2. Estimation of EDR from 1D Intersections of the Turbulent Velocity Field

Atmospheric airborne measurements deliver information on 1D intersections of the turbulent velocity field along plane tracks. From this, under the assumption of local isotropy and with the Taylor's frozen eddy hypothesis, information on turbulence statistics are extracted.

Two commonly used methods that follow from the the Kolmogorov's second similarity hypothesis [9] are the frequency, or wavenumber spectrum and the structure-function approach. The one-dimensional frequency spectra of the longitudinal and transverse velocity components in the inertial range are given by [17]:

$$S_\parallel(f_1) = C_K \left(\frac{U}{2\pi}\right)^{2/3} \epsilon^{2/3} f_1^{-5/3}, \quad S_\perp(f_1) = C_K' \left(\frac{U}{2\pi}\right)^{2/3} \epsilon^{2/3} f_1^{-5/3}. \tag{6}$$

Here $f_1 = U\kappa_1/(2\pi)$ and $\kappa_1$ is the longitudinal component of the wavenumber vector, $C_K \approx 0.49$, and $C_K' \approx 0.65$, $U$ stands for the true air speed (the magnitude of the vector difference between the aircraft velocity and the wind velocity). EDR is estimated from Equation (6) by fitting a line with $-5/3$ slope on a logarithmic plot within a certain range of frequencies, further called the "fitting range", see left panel of Figure 1. The log of the intercept is equivalent to

$$C_K \left(\frac{U}{2\pi}\right)^{2/3} \epsilon^{2/3} \quad \text{or} \quad C_K' \left(\frac{U}{2\pi}\right)^{2/3} \epsilon^{2/3},$$

respectively.

Alternatively, one can consider the second or the third order structure functions. As reliable calculation of the latter demands large sample, which is often not accessible from airborne measurements, the considerations will be restricted to the second order longitudinal or transverse structure functions defined in the inertial range, respectively as

$$D_\parallel(r) = \langle (u_l(x+r,t) - u_l(x,t))^2 \rangle = C_2 \epsilon^{2/3} r^{2/3}, \tag{7}$$

$$D_\perp(r) = \langle (u_n(x+r,t) - u_n(x,t))^2 \rangle = C_2' \epsilon^{2/3} r^{2/3}, \tag{8}$$

here $u_l$ and $u_n$ are, respectively the longitudinal and transverse component of velocity and $r$ is a displacement along the direction defined by $u_l$. We apply values $C_2 \approx 2.$ and $C_2' \approx 2.86$. Analogously, EDR is estimated by fitting a line with a $2/3$ slope on a logarithmic plot, see right panel of Figure 1.

In the previous works [10,11] alternative approaches to estimate EDR from low or moderate-resolution velocity time series, based on zero-crossings and variance of velocity derivative were proposed. The zero-crossings is calculated simply as the number of times the signal crosses the level zero per unit length or per unit time. When a turbulent signal is analysed, the number of zero-crossings per unit length is related to the Taylor's microscale [23,24]. The first method is based on the scaling of the number of crossings of a filtered signal as a function of cut-off frequency. The formula for longitudinal velocity component reads [10]

$$u_1'^2 N_1^2 - u_i'^2 N_i^2 = 3C_K \left(\frac{U}{2\pi}\right)^{2/3} \epsilon^{2/3} \left(f_1^{4/3} - f_i^{4/3}\right), \tag{9}$$

where $f_i, i = 1, \ldots, n$ are cut-off frequencies that are placed within the inertial range, $u_i', i = 1, \ldots, n$ are standard deviations of the filtered signal and $N_i, i = 1, \ldots, n$ is calculated as the number of times the investigated, filtered signal crosses the level zero per unit time. If the transverse velocity component is investigated, $C_K$ in the Formula (9) should by replaced by $C_K'$. It was assumed in (9) that the filter is perfectly rectangular in the Fourier space. Analogously to the previous methods, the value of EDR can be estimated from Equation (9) by a proper curve-fitting, see the left panel of Figure 2.

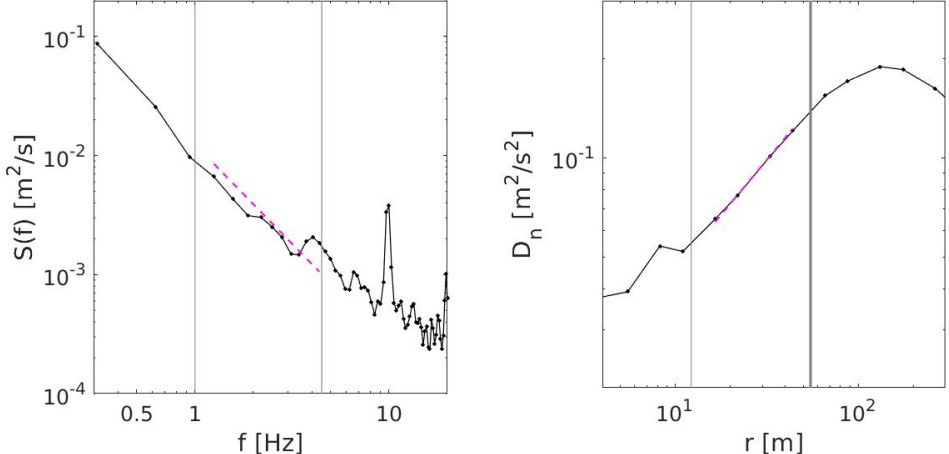

**Figure 1. Left**: Exemplary frequency spectra of the transverse velocity component from experimental POST data [15], **Right**: Exemplary second-order structure function. Vertical lines indicate bounds of the fitting range, magenta dashed line is a curve-fit within this range.

Yet another method proposed in Refs. [10,11] is based on an iterative procedure. The key idea is to recover the part of the spectrum which is missing due to unsufficient resolution or measurement errors using Equation (4) with a prescribed, analytical form of function $f(\eta\kappa)$, see the right panel of Figure 2. Under the local isotropy assumption the EDR can be found from the variance of the derivative of the measured, longitudinal fluctuating velocity component $u'_{cut}$, having the spectral cut-off at $\kappa_{cut}$

$$\epsilon = C_\lambda \nu \left\langle \left( \frac{\partial u'_{cut}}{\partial x} \right)^2 \right\rangle \mathcal{C}_\mathcal{F}, \tag{10}$$

where the constant $C_\lambda = 15$ for the longitudinal and $C'_\lambda = 15/2$ for the transverse velocity component. Above, $\mathcal{C}_\mathcal{F}$ is a correcting factor which represents the unresolved part of the dissipation spectrum. Alternatively, from the relation of [23]

$$\langle (\partial u'/\partial x)^2 \rangle = \langle u'^2 \rangle \pi^2 N^2, \tag{11}$$

the EDR can be expressed in terms of the number zero-crossings $N_{cut}$ per unit length of the filtered signal

$$\epsilon = C_\lambda \pi^2 \nu \langle u'^2_{cut} \rangle N^2_{cut} \mathcal{C}_\mathcal{F}. \tag{12}$$

For a filter perfectly rectangular in the Fourier space we have

$$\left\langle \left( \frac{\partial u'_{cut}}{\partial x} \right)^2 \right\rangle = \int_0^{\kappa_{cut}} \kappa_1^2 E_{11} \mathrm{d}\kappa_1, \tag{13}$$

while for the fully-resolved signal $u'$

$$\left\langle \left( \frac{\partial u'}{\partial x} \right)^2 \right\rangle = \int_0^{\kappa_{cut}} \kappa_1^2 E_{11} \mathrm{d}\kappa_1 + \int_{\kappa_{cut}}^\infty \kappa_1^2 E_{11} \mathrm{d}\kappa_1, \tag{14}$$

where $E_{11}(\kappa_1)$ is the one-sided 1D velocity spectrum. The ratio of Equations (13) and (14) gives the correcting factor $\mathcal{C}_\mathcal{F}$. Unmeasured part of the spectral function $E_{11}(\kappa_1)$ for $\kappa_1 \geq \kappa_{cut}$ is replaced by an analytical formula with the use of Equation (4) and a relationship between the one-dimensional

spectra $E_{11}$ and the spectral energy density $E(\kappa)$. Finally, the formula for $\mathcal{C}_{\mathcal{F}}$ for longitudinal velocity components reads, see Ref. [11]

$$\mathcal{C}_{\mathcal{F}} = 1 + \frac{\int_{k_{cut}\beta\eta}^{\infty} \xi_1^2 \int_{\xi_1}^{\infty} \xi^{-8/3} f_\eta(\xi) \left(1 - \frac{\xi_1^2}{\xi^2}\right) d\xi d\xi_1}{\int_0^{k_{cut}\beta\eta} \xi_1^2 \int_{\xi_1}^{\infty} \xi^{-8/3} f_\eta(\xi) \left(1 - \frac{\xi_1^2}{\xi^2}\right) d\xi d\xi_1}, \tag{15}$$

while for the transverse component we have

$$\mathcal{C}_{\mathcal{F}}' = 1 + \frac{\int_{k_{cut}\beta\eta}^{\infty} \xi_1^2 \int_{\xi_1}^{\infty} \xi^{-8/3} f_\eta(\xi) \left(1 + \frac{\xi_1^2}{\xi^2}\right) d\xi d\xi_1}{\int_0^{k_{cut}\beta\eta} \xi_1^2 \int_{\xi_1}^{\infty} \xi^{-8/3} f_\eta(\xi) \left(1 + \frac{\xi_1^2}{\xi^2}\right) d\xi d\xi_1}. \tag{16}$$

The function $f_\eta$ above is given analytically by the Pope's model [17] for the dissipation range

$$f_\eta(\kappa\eta) = e^{-\beta\left\{\left[(k\eta)^4 + c_\eta^4\right]^{1/4} - c_\eta\right\}}, \tag{17}$$

where $\beta = 5.2$, $c_\eta = 0.4$. For this function the best fit with data from numerical experiment was observed [11].

To estimate $\mathcal{C}_{\mathcal{F}}$, it is necessary to calculate integrals with $\eta$ in the integration bounds. As the Kolmogorov lengthscale $\eta$ depends on $\epsilon$, see Equation (3), an iterative procedure is used to find the final value of EDR. With a first guess of $\epsilon$ the correcting factor $\mathcal{C}_{\mathcal{F}}$ is calculated and introduced into (10) to calculate new value of $\epsilon$. The procedure can be continued until the condition $\Delta\eta = |\eta^{n+1} - \eta^n| < d_\eta$ for a given error value $d_\eta$ is satisfied. Interestingly, independent of the initial guess, results converge very fast to the sought value of $\epsilon$, see Refs. [10,11].

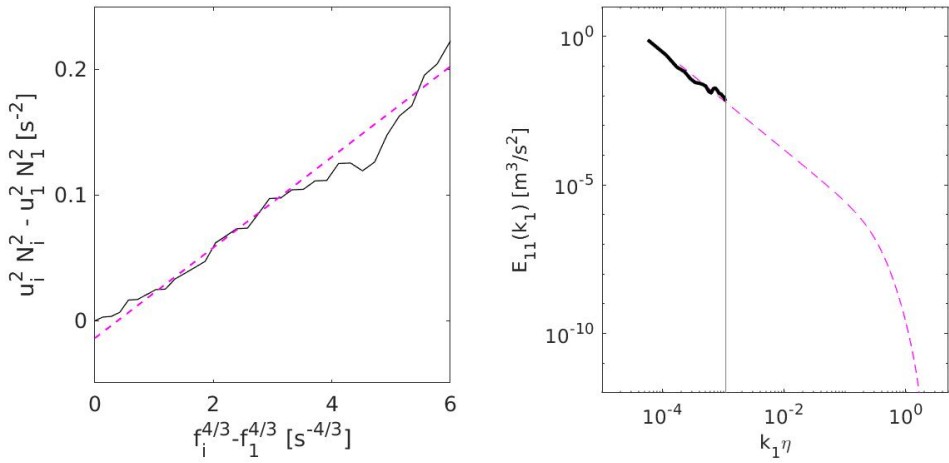

**Figure 2. Left**: Exemplary number of zero-crossing scaling calculated from experimental POST data, for cut-off within the inertial range, see Equation (9), dashed magenta line indicates the best fit-line. **Right**: Power spectra of a signal with spectral cut-off, magenta line indicates reconstructed part of the spectrum.

The preliminary tests of the new method for EDR estimation were performed in Ref. [10] on the data from Stratocumulus top measurement campaign (POST), (see Refs. [12,13,15]) and in Ref. [11] on the numerical Direct Numerical Simulation (DNS) data of stratocumulus cloud-top. The results show a good agreement with standard EDR estimates from Equation (5). A clear advantage of the iterative methods is that, as the Formula (4) is universal, only the value of the effective spectral cut-off

$\kappa_{cut}$ should be given to estimate $\mathcal{C}_\mathcal{F}$ and $\epsilon$ from Equation (10). Additionally, it was argued in Ref. [10] that the new methods respond differently to errors due to finite sampling and finite averaging window than the standard spectral methods, e.g., they are less sensitive to the bias error. This, particularly for the case of small sampling frequencies increases robustness of $\epsilon$ estimates. Within the present paper we perform a more detailed study of the performance of all the above-described methods, first on artificial and next on real signals.

*2.3. External Intermittency*

It was argued in Ref. [11] that the number of zero- crossings statistics can be used as an indicator of external or global intermittency, that is, the existence of of laminar spots within the turbulent flow. Such situation can be present e.g., at the top of a boundary layer, as intermittent turbulence-driven bursts into the free troposphere [25] or the top of stratocumulus cloud due to entrainment of the cloud-free, laminar air from above the cloud. Moreover, the presence of stable stratification which damps turbulent motions can lead to partial re-laminarisation of the flow. The latter mechanism is observed also in large-scale oceanic motions where the turbulence is anisotropic and of intermittent structure [26,27]. Large scale intermittency of vertical velocity component was reported in Ref. [28] for stratified turbulence in a distinct range of Froude numbers. A rough detection of external intermittency from measurement data may help to better describe and quantify atmospheric and oceanic turbulence processes.

Detection of external intermittency usually involves a proper choice of an indicator function $q$, a criterion function $f(q)$ and a threshold level $T_h$. The flow is assumed turbulent when $f(q) > T_h$ [29]. The time fraction when turbulent flow is detected is called the intermittency factor $\gamma$. It takes value 1 if the flow is fully turbulent, 0 if it is laminar and $0 < \gamma < 1$ if it is intermittent. Examples of $q$ are the velocity derivative for 1D signals, enstrophy [16] when full 3D velocity field is known or density in the case of stratified flows [30,31]. Another approach was proposed in Ref. [11]. Therein, it was assumed that in the externally intermittent flow, the statistics will change to $\gamma\langle u^2\rangle$ and $\gamma\langle(\partial u/\partial x)^2\rangle$. Moreover, as the laminar part of the signal does not significantly contribute to the number of crossings, hence, $\gamma N_L$ crossings per unit length will be detected in the intermittent signal. In such approximation, the Taylor microscale (here, we consider the transverse microscale) will remain unchanged in the intermittent flow

$$\lambda_{nI} = \left[\frac{\gamma\langle u'^2\rangle}{\gamma\langle(\partial u'/\partial x)^2\rangle}\right]^{1/2} = \lambda_n, \tag{18}$$

where the subscripts $I$ are related to the statistics in the intermittent flow. On the other hand, the length scale defined based on zero-crossings, the so-called Liepmann scale is modified to

$$\Lambda_I = \frac{1}{\gamma\pi N_L} = \frac{1}{\gamma}\Lambda \quad \text{hence,} \quad \frac{\lambda_{nI}}{\Lambda_I} = \gamma\frac{\lambda_n}{\Lambda}. \tag{19}$$

This suggests that the ratio of the transverse Taylor to Liepmann length scales is an indicator of the external intermittency. This also imposes, in case of external intermittency, the EDR estimated from Equation (12) will be underpredicted in comparison to estimates based on power spectrum, structure function or the variance of velocity derivative. If, on the other hand the $\lambda_n/\Lambda$ ratio oscillates around a value close to unity along the investigated signal, we can expect the flow is fully turbulent. Within this work we calculate the $\lambda_n/\Lambda$ ratio for artificial signals and compare it with the prescribed value of the intermittency parameter $\gamma$.

**3. Error Analysis of EDR Estimates**

Due to the limitations of measuring sensors and rapidly changing weather conditions, determination of $\epsilon$ from aircraft measurements is subject to errors, among others, due to the finite frequency of measurements and a short averaging window. They cause modification of the energy spectra and difficulties in estimating EDR on their basis. Finite windowing of the measured signal leads

to a spectral leakage, i.e., the spectral energy connected with a given frequency "leaks" to different, adjacent frequencies.

For the signal to be fully resolved the sampling frequency should be at least equal to twice the reciprocal of the Kolmogorov's time scale $2/\tau$. In practice, due to limitation of sensors, velocity in airborne measurements is sampled with frequency $f_s$ which is smaller by 3–4 orders of magnitude. This under-sampling creates aliases of the true spectrum [32–34] such that to the Fourier transform of a sampled signal its $n$-th harmonics are added. With this, the spectral energy density of turbulence kinetic energy deviates from the Kolmogorov's $-5/3$ scaling within a certain range of frequencies. In most applications, the region of spectrum most influenced by aliasing is skipped and the calculated spectrum is fitted to the Kolmogorov's formula in a reduced range of frequencies. This, as a consequence, increases uncertainty of EDR estimates. Moreover, the optimal choice of the fitting range is not obvious and highly case-dependent.

To test sensitivity of EDR retrieval methods we generated artificial velocity time series based on the von Karman model spectrum [33,35,36] model, such that the one-sided frequency spectra of the signals have a prescribed form

$$S(f) \approx C \, \frac{2\pi}{U} \, \frac{u'^2 L_0}{\left[1 + L_0^2 \left(\frac{2\pi f}{U}\right)^2\right]^{5/6}}, \tag{20}$$

where $C = C_K \approx 0.49$ was assumed. Coefficients of the Fourier series expansion of velocity signal were calculated as

$$w_j = \sqrt{W_j}(a + ib), \tag{21}$$

where $i = \sqrt{-1}$, $a$ and $b$ are random numbers from the standard Gaussian distribution with zero mean and unit variance and $W_j = S(f_j)\Delta f$, $j = 1, \ldots, N$. To test the proposed methods for EDR retrieval we used $L_0 = 10$ m, $U = 55$ ms$^{-1}$ and $u' = 0.18$ ms$^{-1}$ in Equation (20). The length of the signal was $N = 2^{16}$ points and the sampling frequency was $f_s = 200$ Hz. The reference EDR calculated from the theoretical profile (20) was $\epsilon_{ref} = 2.97 \times 10^{-4}$ m$^2$/s$^3$. We used this value to non-dimensionalze results of EDR retrieval for the power spectra, number of crossings and velocity variance method. The artificially generated signals may not reproduce all statistics of turbulent field correctly (this problem was also mentioned in Ref. [36]), with (20) the EDR estimated from the structure function were slightly different even for very large averaging windows, hence results for structure function were non-dimensionalized with the slightly different $\epsilon_{ref} = 2.49 \times 10^{-4}$ m$^2$/s$^3$. A sample signal is given at the upper plots in Figure 3.

In the next step, we considered real signals from the flight 13 of the CIRPAS Twin Otter research aircraft, collected with sampling frequency $f_s = 40$ Hz during the POST campaign [12–14]. In the investigated horizontal segment atmospheric conditions remained approximately constant, the true air speed (the magnitude of the vector difference between the aircraft velocity and the wind velocity), was $U = 55$ ms$^{-1}$, the turbulence intensity $u' = 0.35$ ms$^{-1}$, the sampling frequency $f_s = 40$ Hz and the length of the segment was $N = 19,817$ points, which corresponds to the total time of 495 s. Here, for $\epsilon_{ref}$ (the same for all methods, including the structure function approach) we have chosen estimation from the power spectra (6) with statistics averaged over the whole length of the signal $\epsilon_{ref} = 7.96 \times 10^{-4}$ m$^2$/s$^3$.

To calculate EDR, the size of averaging window should first be estimated. It should be larger than the turbulence characteristic eddy turn-over time $T_{ch}$. We assume this time scale equals $T_{ch} = U/L$, where L and U are, respectively, the characteristic length and velocity scales of turbulence, estimated as in the $\mathcal{K} - \epsilon$ turbulence model [17]

$$L = C_\mu \frac{\mathcal{K}^{3/2}}{\epsilon}, \quad U = \sqrt{\mathcal{K}} = \sqrt{\frac{3}{2}} u', \quad T_{ch} = \frac{L}{U}. \tag{22}$$

Above, $C_\mu = 0.09$ is the model constant. This length scale is around 10 times smaller than the integral length scale $\mathcal{L}$ from Equation (2), as $C_\epsilon \approx 10C_\mu$. With the fixed size of the averaging window, running averages are calculated along the signal and EDR is approximated based on the power spectrum method (6), the structure function (7), number of crossings scaling in the inertial range (9), and the iterative methods based on the velocity variance (10) and the number of crossings (12). The estimates are denoted, respectively, as $\epsilon_{PS}$, $\epsilon_{SF}$, $\epsilon_{NCF}$, $\epsilon_{VAR}$ and $\epsilon_{NCR}$. For the investigated, artificial signals $T_{ch} = 9.84$ s which, for the frequency $f_s = 200$ Hz corresponds to 2000 points, approximately. Sample results of EDR estimates for the averaging window $T_{ch}/4$ and $T_{ch}$ are presented in the middle and bottom plots in Figure 3. It is observed, the latter results are obviously smoother. Here, the fitting range $f = [10 \div 20]$ Hz and the cut-off frequency for the iterative methods was $f_{cut} = 20$ Hz. In the tests we calculated the mean value and the standard deviation of such estimates for averaging windows ranging from $T_{ch}/4$ to $2T_{ch}$. Moreover, we reduced frequency of the signals by successive downsampling. We also investigated influence of results for different choices of the fitting range and the cut-off frequency. The filtering of the signal was performed with the the sixth order low-pass Butterworth filter [37] implemented in Matlab ®.

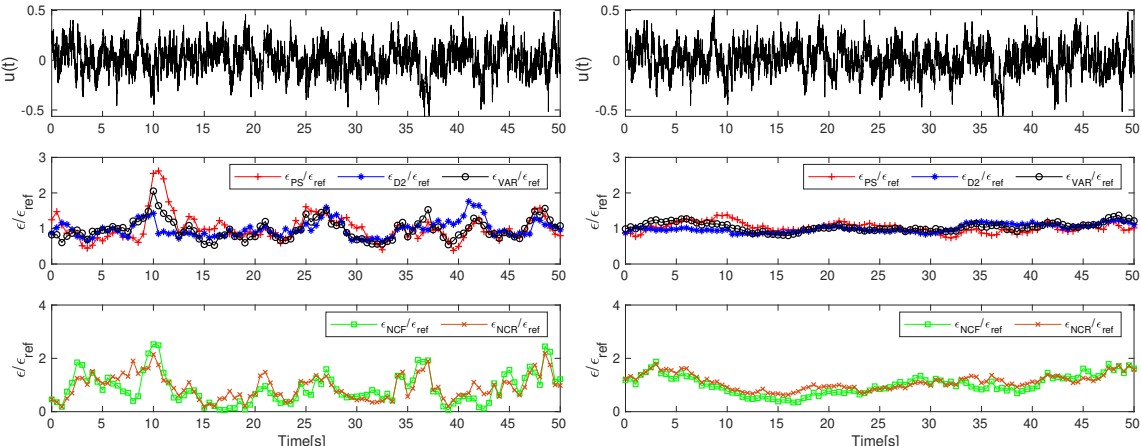

**Figure 3.** Top plots: Part of an artificial signal, middle plots: EDR estimates from the power spectrum $\epsilon_{PS}$, Equation (6), structure function $\epsilon_{SF}$, Equation (7) and number of crossings scaling in the inertial range $\epsilon_{NCF}$, Equation (9), bottom plots: EDR estimates from the iterative method based on the velocity variance $\epsilon_{VAR}$, Equation (10) and the number of zero-crossings per unit length, Equation (12). Left column: averaging window $T_{ch}/4$, right column: averaging window $T_{ch}$.

## 4. EDR Retrieval from Artificial Signals

### 4.1. Different Averaging Windows

In the first test we keep the sampling frequency $f_s = 200$ Hz constant and vary the size of averaging window. For this frequency, as the characteristic time scale $T_{ch} \approx 9.84$ s, the size of averaging window changes from $2T_{ch}$, which corresponds to, approximately $N_{av} = 4000$ signal points to $T_{ch}/4$ which corresponds to approximately 500 points. The signal was fitted first within the range $f = [10 \div 20]$ Hz in case of $\epsilon_{PS}$, $\epsilon_{SF}$, $\epsilon_{NCF}$ estimates or low-passed filtered with the cut-off frequency $f_{cut} = 20$ Hz in case of iterative methods for $\epsilon_{VAR}$ and $\epsilon_{NCR}$. Results are compared in Figure 4. It is seen, the value of $\epsilon_{SF}$, averaged along the signal is the least and the mean $\epsilon_{PS}$ the most affected by the decrease of the size of the averaging window. The mean of $\epsilon_{VAR}$ also performs well in this test. The right panel presents standard deviations of the estimates from their mean value The highest deviations are observed for the methods based on the number of crossings, which is in line with preliminary tests performed in Refs. [10,11] for $\epsilon_{NCF}$. Next, the fitting range was changed by moving it towards larger scales. Estimates for $f = [5 \div 10]$ Hz and $f = [1 \div 5]$ Hz are presented in Figures 5 and 6, respectively. As it is seen, estimates from the second-order structure function become under-predicted.

Other results remain comparable, also the difference of standard deviations of $\epsilon_{NCF}$ and remaining methods is smaller for $f = [1 \div 5]$ Hz.

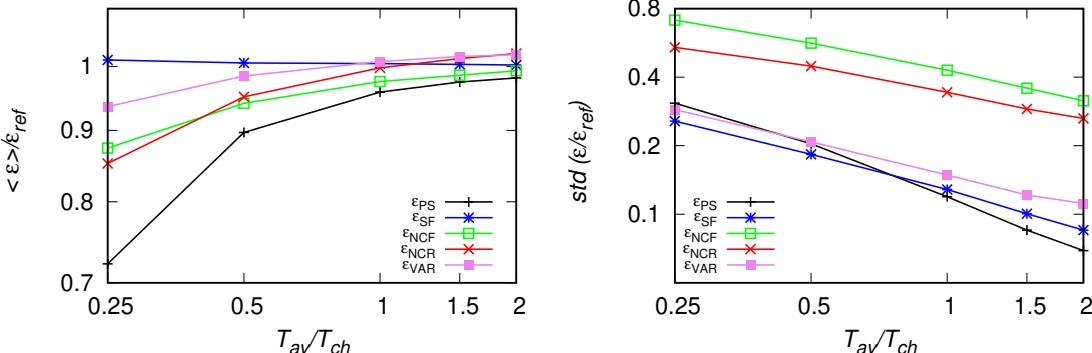

**Figure 4.** Statistics of EDR estimates for the fitting range $f = [10 \div 20]$ Hz, or $f_{cut} = 20$ Hz from artificial signals with $f_s = 200$ Hz. **Left panel**: EDR averaged along the signal, **right panel**: standard deviation of the estimates. EDR estimates based on the power spectrum $\epsilon_{PS}$, Equation (6), structure function $\epsilon_{SF}$, Equation (7), number of zero-crossings scaling in the inertial range $\epsilon_{NCF}$, Equation (9), iterative method based on the velocity variance $\epsilon_{VAR}$, Equation (10) and the number of zero-crossings per unit length, Equation (12).

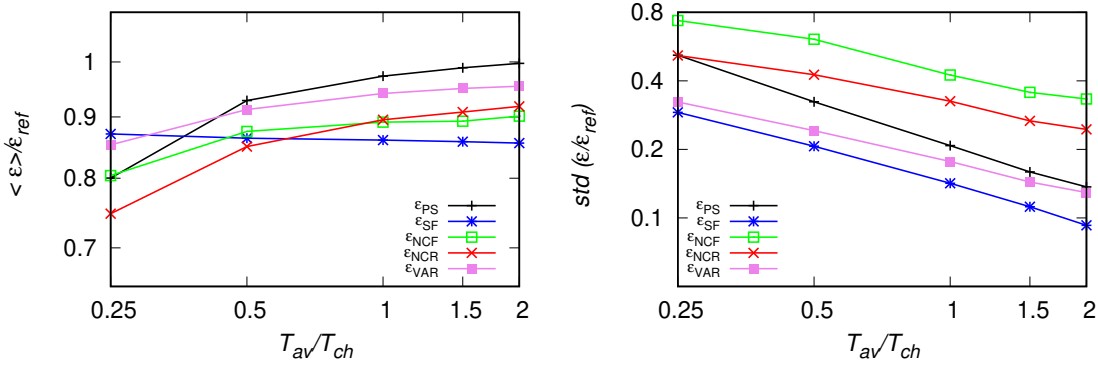

**Figure 5.** As in Figure 4, but for the fitting range $f = [5 \div 10]$ Hz.

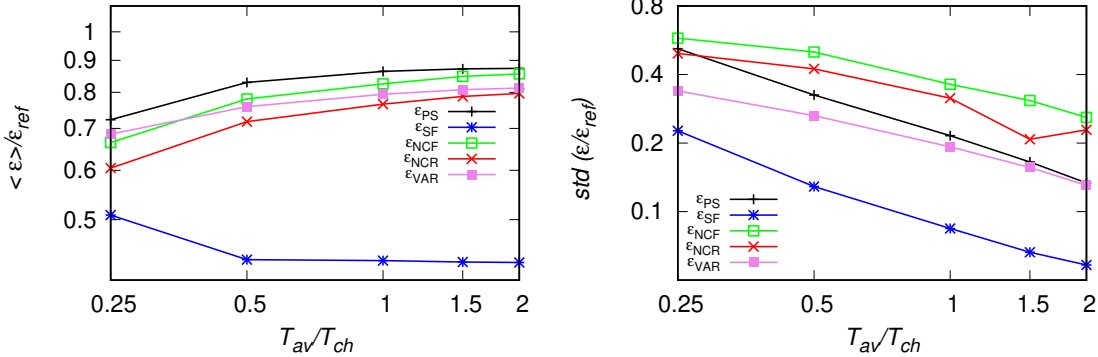

**Figure 6.** As in Figure 4, but for the fitting range $f = [1 \div 5]$ Hz.

### 4.2. Different Averaging Windows and Different Sampling Frequencies

Next, we investigated in detail how different methods respond to changes of both, averaging window and the sampling frequency. For this we successively downsampled the signal. For the fitting range $f = [10 \div 20]$ Hz it was done twice, to create signals with $f_s = 100$ Hz and $f_s = 50$ Hz. The cut-off frequency in the iterative method was $f_{cut} = 20$ Hz. The same tests were repeated for the fitting range $f = [5 \div 10]$ Hz or $f_{cut} = 10$ Hz and in this case the downsampling could have been performed three times, down to $f_s = 25$ Hz. Finally, calculations for the fitting range $f = [1 \div 5]$ Hz or $f_{cut} = 5$ Hz and $f_s = 200$ Hz, 100 Hz, 50 Hz and 12.5 Hz were performed. We also note in passing that the moving window of a fixed size, e.g., $T_{ch}$ correspond to 2000 signal points in case of $f_s = 200$ Hz, to 1000 points for $f_s = 100$ Hz, and to 500 points in case of $f_s = 50$ Hz.

#### 4.2.1. Power Spectra

First, results of $\epsilon_{PS}$ estimated from the power spectra, Equation (6), are presented in Figures 7–9. It is observed, the mean value of $\epsilon_{PS}$ (averaged along the signal) becomes over-predicted with decreasing sampling frequency. This bias error follows from the aliasing (see discussion in Section 3) which leads to deviation from the Kolmogorov's $-5/3$ scaling, especially in the high-wavenumber part of the spectrum. Standard deviation of $\epsilon_{PS}$, presented in the right panels in Figures 7–9 is affected mainly by the size of averaging window. Results become more scattered for smaller $T_{av}$.

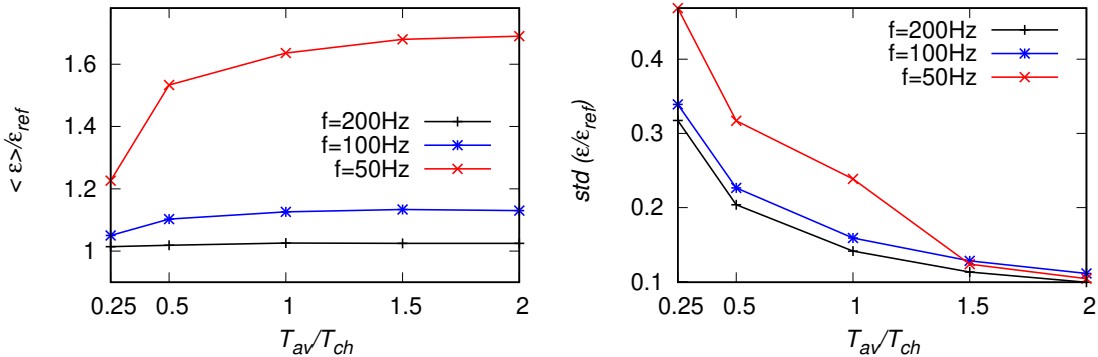

**Figure 7.** Statistics of $\epsilon_{PS}$ estimates based on the power spectra, Equation (6), for the fitting range $f = [10 \div 20]$ Hz, from artificial signals with $f_s = 200$ Hz, 100 Hz and 50 Hz. **Left panel**: EDR averaged along the signal, **right panel**: standard deviation of the estimates.

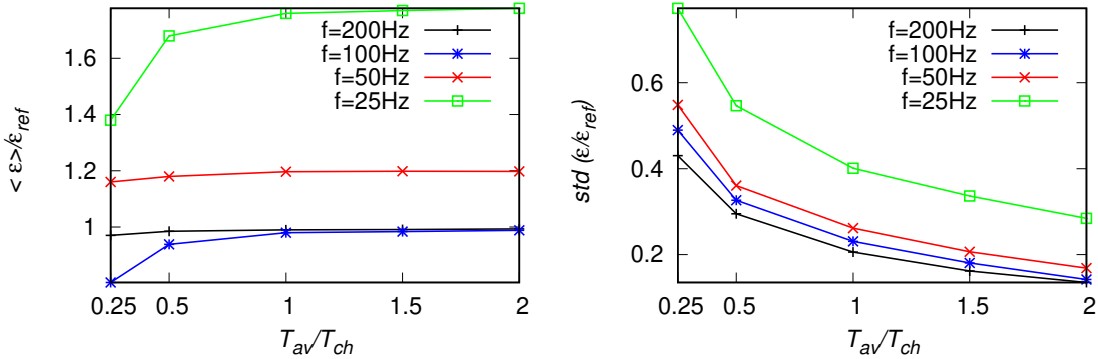

**Figure 8.** As in Figure 7, but for the fitting range $f = [5 \div 10]$ Hz.

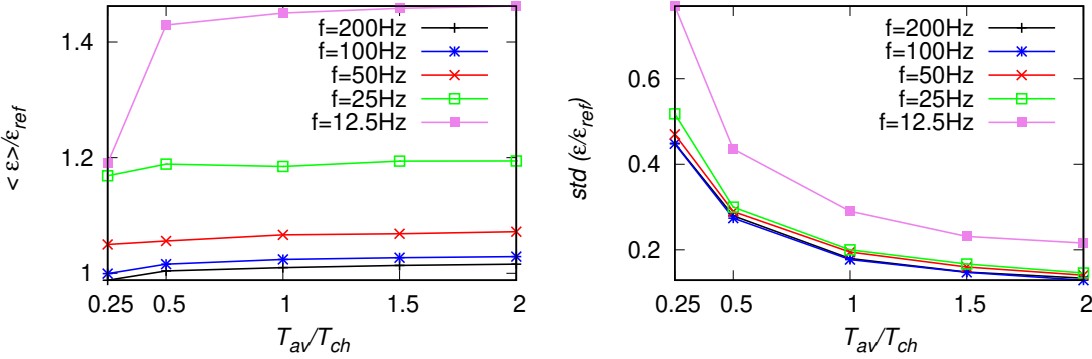

**Figure 9.** As in Figure 7, but for the fitting range $f = [1 \div 5]$ Hz.

### 4.2.2. Structure Functions

Next, results of estimates from the second-order structure function, Equation (7), are presented in Figures 10–12. Interestingly, the mean values (left panels) are hardly influenced by the size of averaging window or the sampling frequency, but they become under-predicted when the fitting range is moved towards smaller wavenumbers (larger scales). The effect of aliasing is smaller and opposite to the one seen in $\epsilon_{PS}$—results decrease with decreasing $f_s$. The standard deviations (right panels in Figures 10–12) are not affected either by the sampling frequency or the choice of the fitting range. Here, the values of standard deviations are smaller than respective results for $\epsilon_{PS}$.

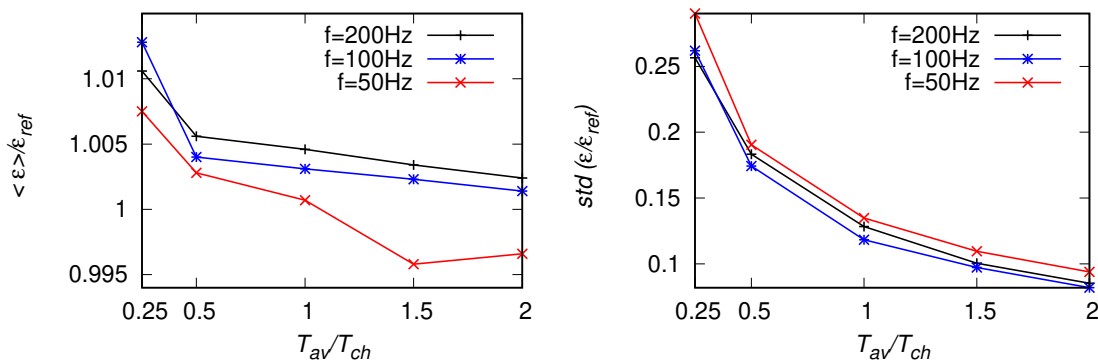

**Figure 10.** Statistics of $\epsilon_{SF}$ estimates based on the second-order structure function, Equation (7), for the fitting range $f = [10 \div 20]$ Hz, from artificial signals with $f_s = 200$ Hz, 100 Hz and 50 Hz. **Left panel**: EDR averaged along the signal, **right panel**: standard deviation of the estimates.

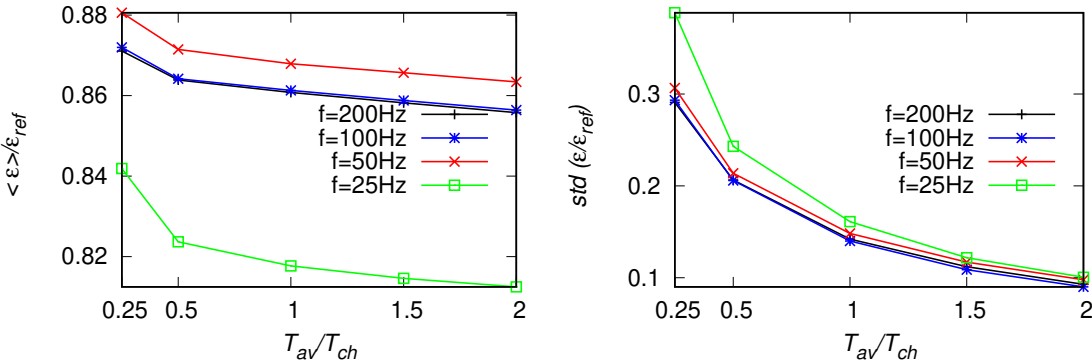

**Figure 11.** As in Figure 10, but for the fitting range $f = [5 \div 10]$ Hz.

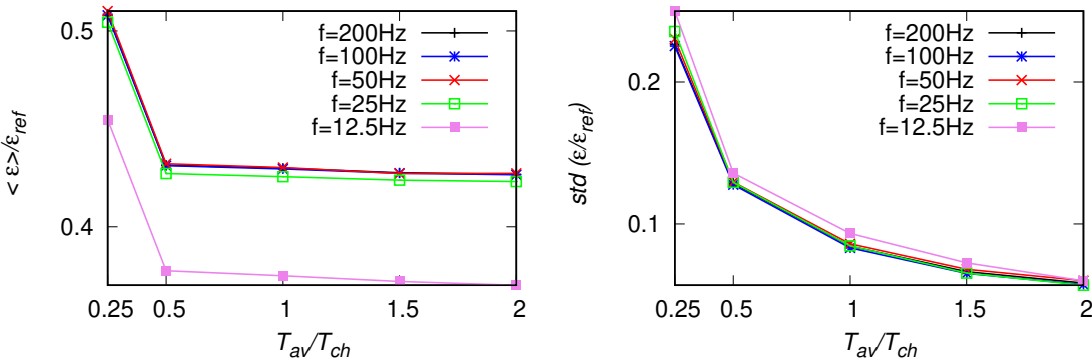

**Figure 12.** As in Figure 10, but for the fitting range $f = [1 \div 5]$ Hz.

### 4.2.3. Number of Zero-Crossings

Results of EDR estimated from the number of zero-crossing scaling in the inertial range, Equation (9), are shown in Figures 13–15. While the sampling frequency decreases, the effect of aliasing becomes visible—the averaged $\epsilon_{NCF}$ become over-predicted, but less than it was observed for $\epsilon_{PS}$. Moreover, the mean values tend to decrease with the decreasing size of the averaging window, as it was observed for $\epsilon_{PS}$. For the mean values, dependency on the fitting range is visible—results become under-predicted (up to about 20%) when the bounds of the range are moved towards smaller wavenumbers. The standard deviations of $\epsilon_{NCF}$ are larger than those of $\epsilon_{PS}$ and $\epsilon_{SF}$.

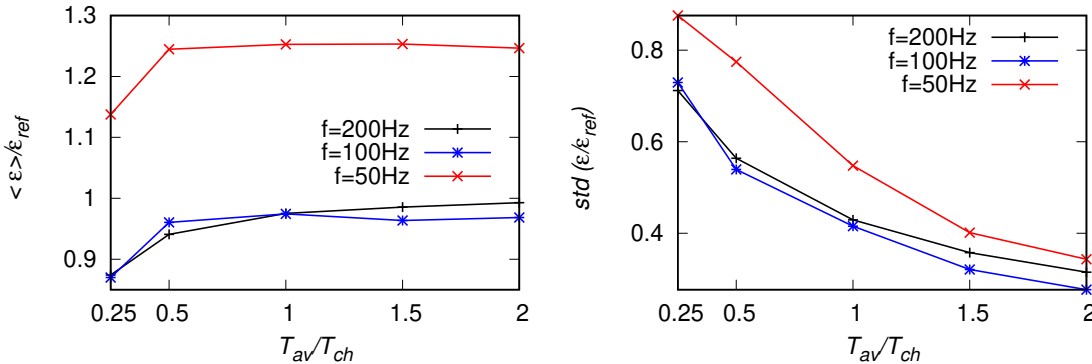

**Figure 13.** Statistics of $\epsilon_{NCF}$ estimates based on the number of zero-crossing scaling, Equation (9), for the fitting range $f = [10 \div 20]$ Hz, from artificial signals with $f_s = 200$ Hz, 100 Hz and 50 Hz. **Left panel**: EDR averaged along the signal, **right panel**: standard deviation of the estimates.

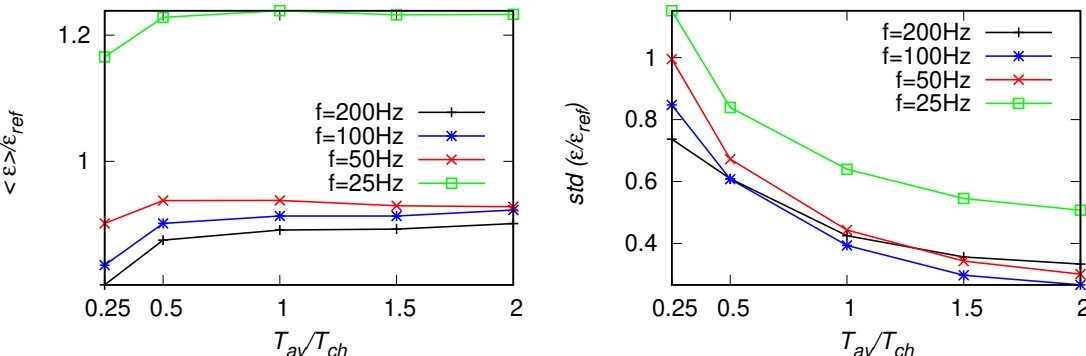

**Figure 14.** As in Figure 13, but for the fitting range $f = [5 \div 10]$ Hz.

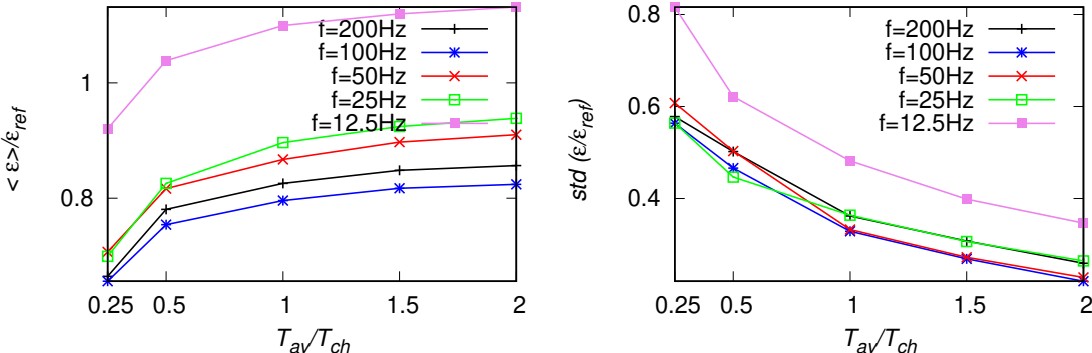

**Figure 15.** As in Figure 13, but for the fitting range $f = [1 \div 5]$ Hz.

#### 4.2.4. Iterative Method

Results for the iterative method $\epsilon_{VAR}$ based on the variance of velocity derivative are presented in Figures 16–18. We omit here $\epsilon_{NCR}$ estimates, as the results are similar to $\epsilon_{NCF}$ (since both are based on the zero-crossing statistics). The mean values presented in the figures differ, at most, by about 20% from the reference value. This is less than maximal deviations observed for the mean values of $\epsilon_{PS}$ and $\epsilon_{SF}$. Moreover, the standard deviations of $\epsilon_{NCR}$ are comparable with those of $\epsilon_{PS}$ and $\epsilon_{SF}$ and smaller than those of $\epsilon_{NCF}$. Smaller bias and the scatter comparable to standard methods suggest the iterative approach could be advantageous. However, when certain setting are considered, e.g., the fitting range of $f = [1 \div 5]$ Hz and $f_s = 50$ Hz, results of $\epsilon_{PS}$ are closer to the reference value than $\epsilon_{VAR}$, which are biased when the fitting range is moved towards smaller frequencies.

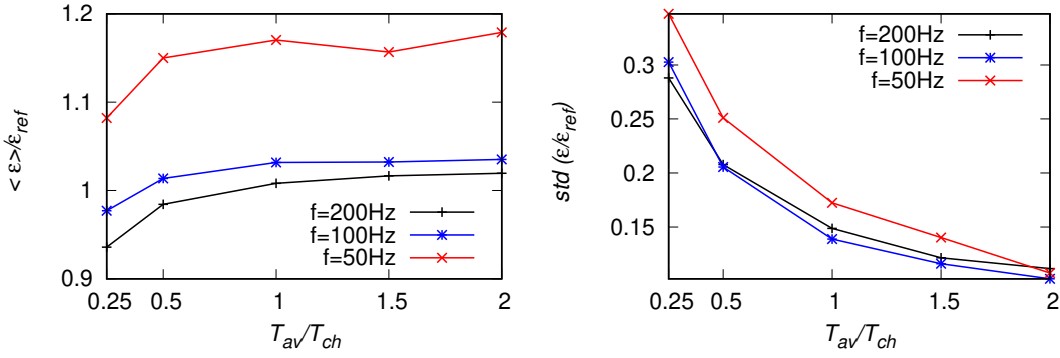

**Figure 16.** Statistics of $\epsilon_{VAR}$ estimates based on the variance of velocity derivative, Equation (10), for the fitting range $f = [10 \div 20]$ Hz, from artificial signals with $f_s = 200$ Hz, 100 Hz and 50 Hz. **Left panel**: EDR averaged along the signal, **right panel**: standard deviation of the estimates.

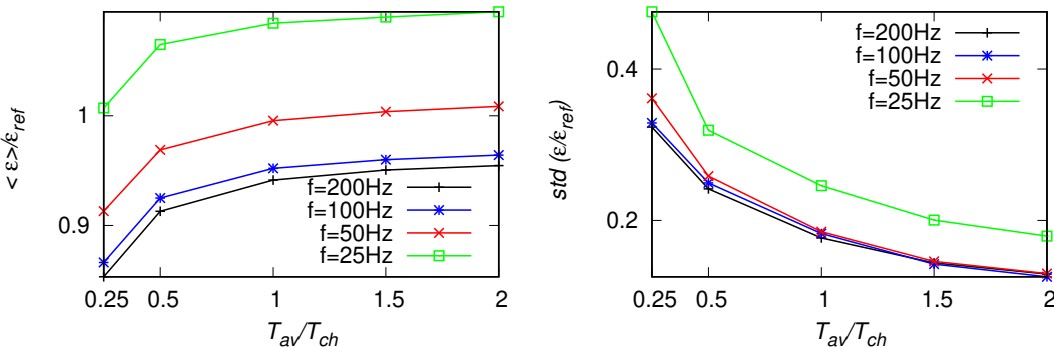

**Figure 17.** As in Figure 16, but for the fitting range $f = [5 \div 10]$ Hz.

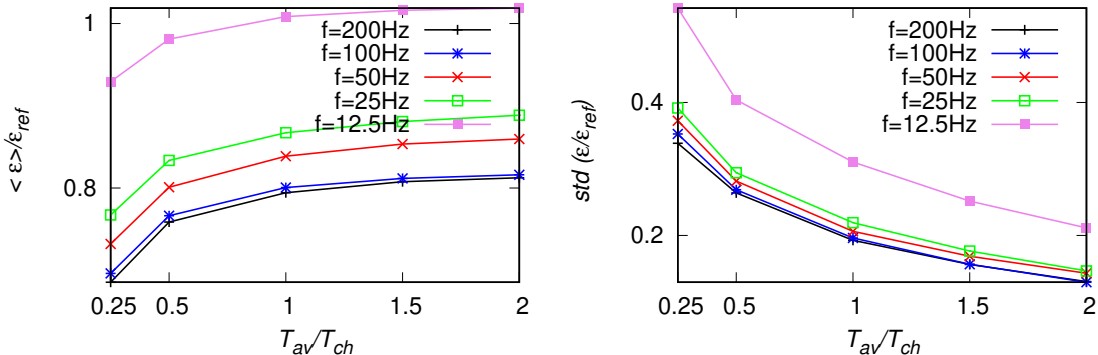

**Figure 18.** As in Figure 16, but for the fitting range $f = [1 \div 5]$ Hz.

### 4.3. Deviations from the Kolmogorov's Scaling

It can be argued that deviations between estimates of different methods indicate problems with the size of averaging window, sampling frequency or the fitting range. However, when analysis of signals from real measurements is performed, differences can exist even in case of sufficiently large $T_{av}$ and sufficiently high $f_s$. This can be a result of deviations from the Kolmogorov's scaling. To investigate this issue we created artificial signals with the scaling larger or smaller than $-5/3 \approx -1.67$, by changing the $5/6$ exponent in the denominator in Equation (20) to smaller/larger values. Mean of EDR estimates for such signals are presented on the log plot in the left panel of Figure 19. As it is observed, $\epsilon_{SF}$ deviates the most from results of other methods. It is larger for scaling exponents smaller than $-5/3$ and smaller for scaling exponent larger than $-5/3$. Results of $\epsilon_{PS}$ and $\epsilon_{NCF}$ are very similar, while $\epsilon_{VAR}$ lies between $\epsilon_{SF}$ and $\epsilon_{PS}$. We also investigated the case of deviations of the Kolmogorov's constant $C_K$ in Equation (6) by changing the constant $C$ in Equation (20). Over-prediction of this value was observed e.g., in Ref. [11] for the spectra of the vertical velocity component. Results of estimates are presented in the right panel of Figure 19. Here, no deviations between the methods are observed.

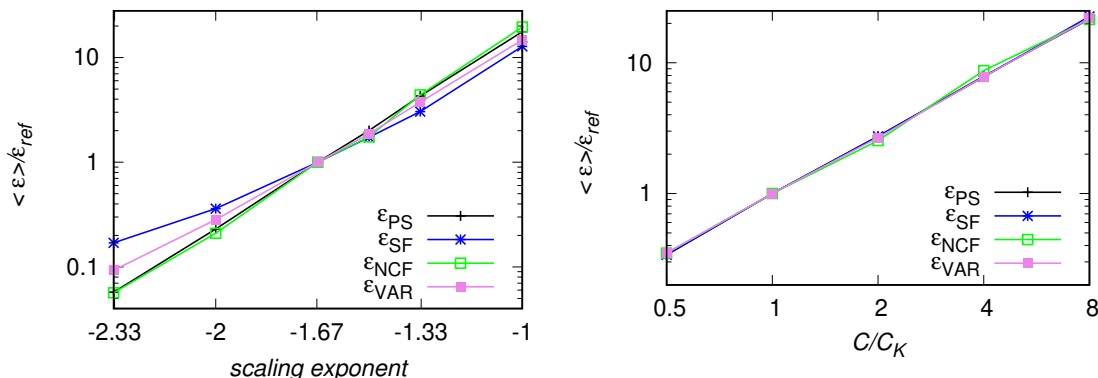

**Figure 19.** Mean of EDR estimates for artificial signals which deviate from the $-5/3$ scaling (left panel) or with deviations from the standard value of the Kolmogorov's constant $C_K \approx 0.49$.

### 5. EDR Retrieval from POST Signals

Next, tests analogous to the ones described in the previous subsection were performed for a signal from the POST campaign [12–15]. The frequency of the original signal was $f_s = 40$ Hz. With $u' = 0.35$ ms$^{-1}$ and $\epsilon_{ref} = 7.96 \times 10^{-4}$ m$^2$/s$^3$ the characteristic time estimated from Equation (22) is $T_{ch} \approx 21$ s which corresponds to the moving window of 1200 points for $f_s = 40$ Hz, 600 points for a signal down-sampled once to $f_s = 20$ Hz and 300 points for the signal down-sampled twice. For the reference value $\epsilon_{ref}$ we chose the mean of $\epsilon_{PS}$, averaged over the whole signal. Results for two different fitting ranges $f = [1 \div 5]$ Hz and $f = [0.2 \div 5]$ Hz are presented in Figures 20–25.

As the lower-bound of the fitting range was the same, results of $\epsilon_{VAR}$ are presented only once, for $f_{cut} = 5$ Hz, see Figure 26. The fact that this method does not depend on the upper-bound of the fitting range is an advantage here. The observed trends are similar as for the artificial signals investigated in Section 4. With decreasing $f_s$ an overprediction of mean value of $\epsilon_{PS}$, up to about 100% is observed in Figures 20 and 21. Moreover, the mean EDR estimates decrease with decreasing size of the averaging window. Smaller over-prediction (about 50%) of $\epsilon_{NCF}$ and $\epsilon_{VAR}$ is observed in Figures 24–26, respectively. On the other hand, the values of mean $\epsilon_{SF}$ increase with decreasing $T_{av}$ and become underpredicted when the upper bound of the fitting range is moved towards larger scales. The highest standard deviation (highest scatter) is observed for $\epsilon_{NCF}$.

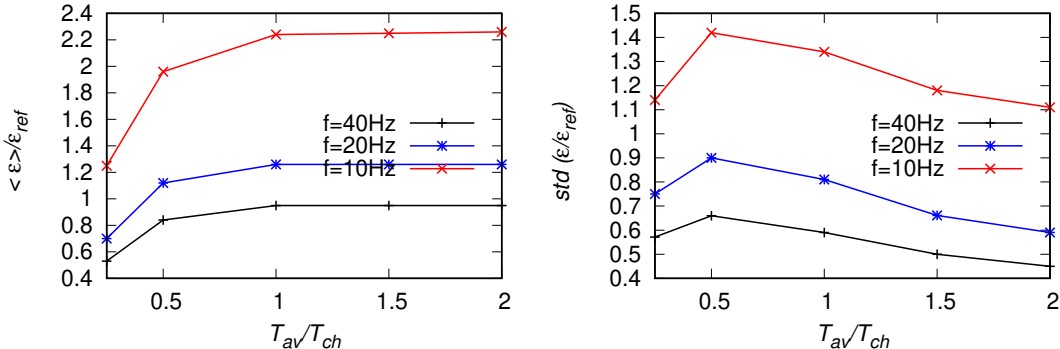

**Figure 20.** Statistics of $\epsilon_{PS}$ estimates based on the power spectra, Equation (6), for the fitting range $f = [1 \div 5]$ Hz, from POST signals [15] with $f_s = 40$ Hz, 20 Hz and 10 Hz. **Left panel**: EDR averaged along the signal, **right panel**: standard deviation of the estimates.

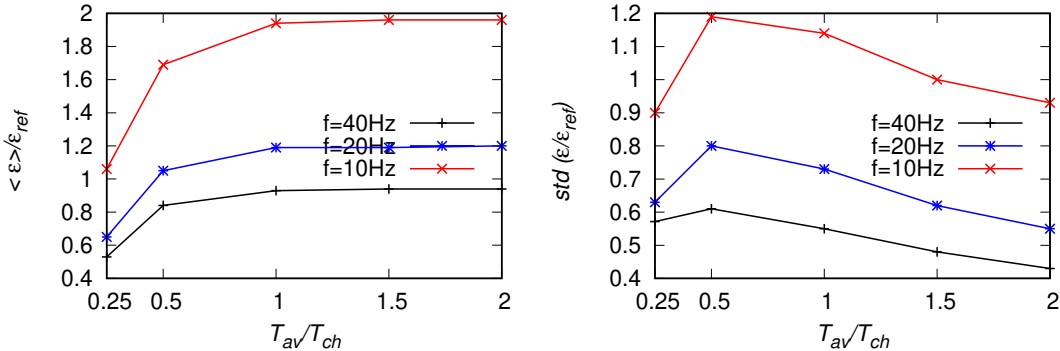

**Figure 21.** As in Figure 20 but for the fitting range $f = [0.2 \div 5]$ Hz.

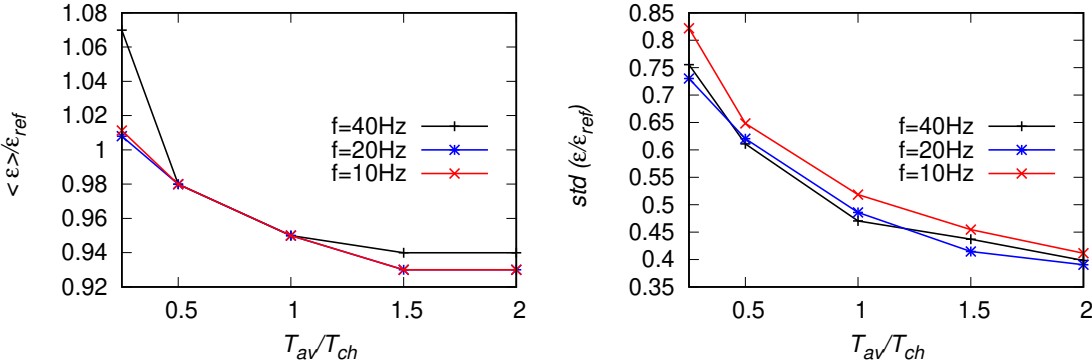

**Figure 22.** Statistics of $\epsilon_{SF}$ estimates based on the structure function, Equation (7), for the fitting range $f = [1 \div 5]$ Hz, from POST signals [15] with $f_s = 40$ Hz, 20 Hz and 10 Hz. **Left panel**: EDR averaged along the signal, **right panel**: standard deviation of the estimates.

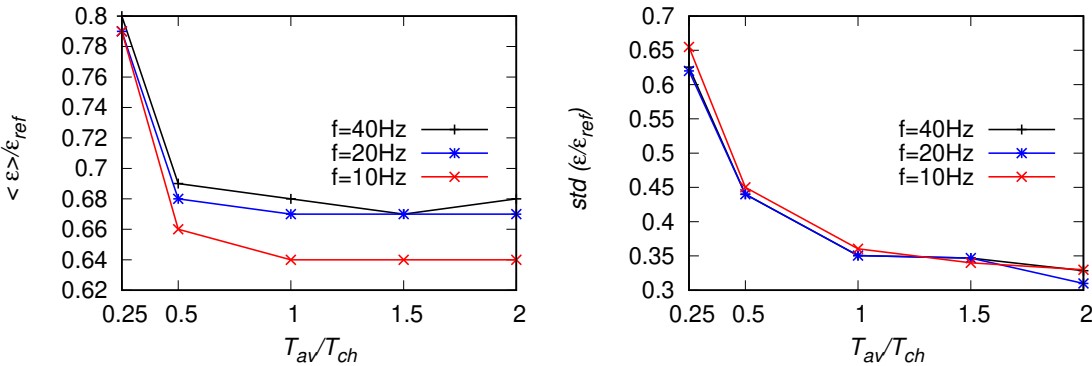

**Figure 23.** As in Figure 22 but for the fitting range $f = [0.2 \div 5]$ Hz.

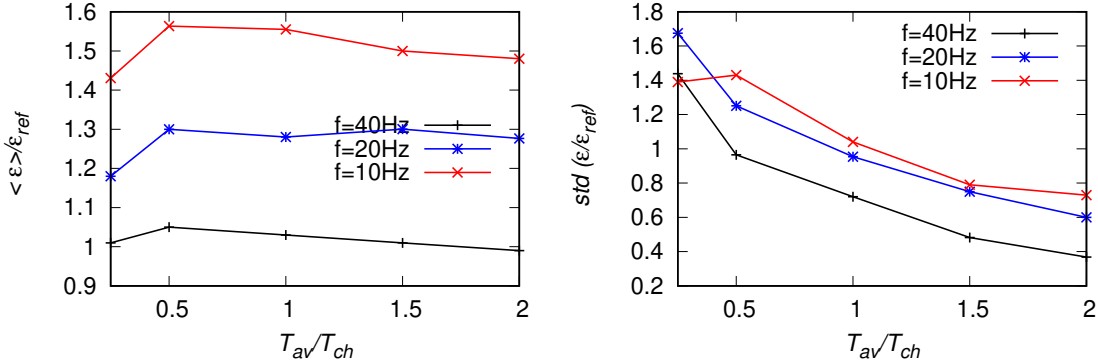

**Figure 24.** Statistics of $\epsilon_{NCF}$ estimates based on the number of zero-crossings, Equation (9), for the fitting range $f = [1 \div 5]$ Hz, from POST signals [15] with $f_s = 40$ Hz, 20 Hz and 10 Hz. **Left panel**: EDR averaged along the signal, **right panel**: standard deviation of the estimates.

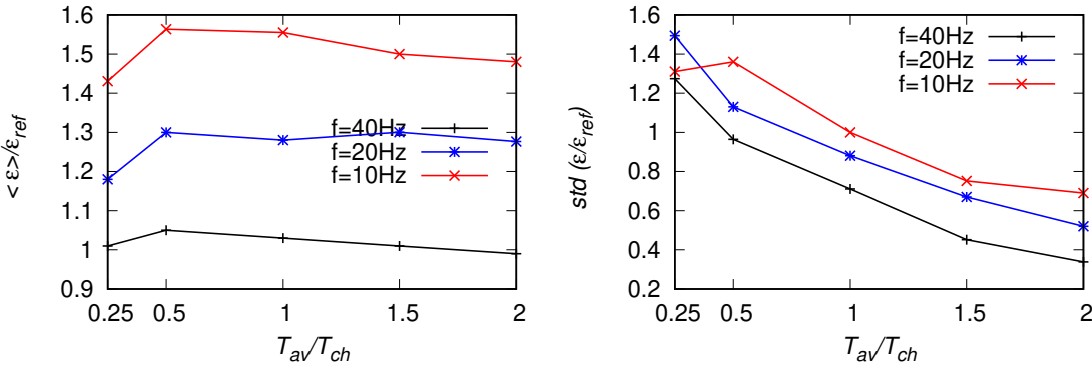

**Figure 25.** As in Figure 24 but for the fitting range $f = [0.2 \div 5]$ Hz.

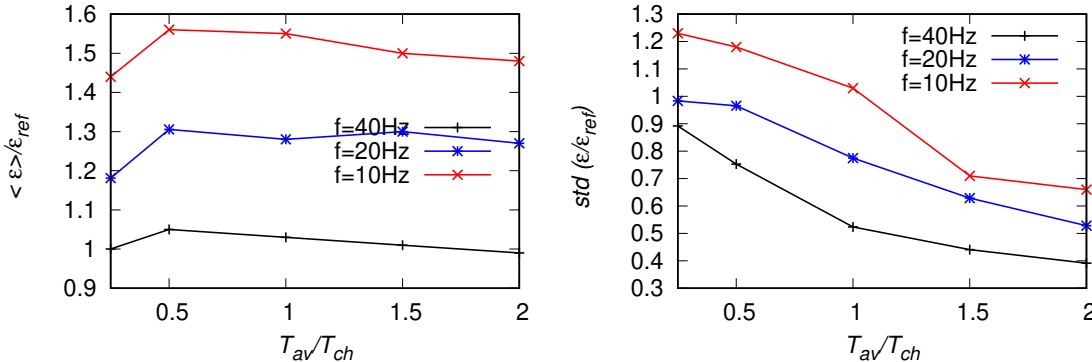

**Figure 26.** Statistics of $\epsilon_{VAR}$ estimates based on the iterative method, Equation (10), for the fitting range $f = [1 \div 5]$ Hz, from POST signals [15] with $f_s = 40$ Hz, 20 Hz and 10 Hz. **Left panel**: EDR averaged along the signal, **right panel**: standard deviation of the estimates.

## 6. Intermittency in Atmospheric Turbulence

The last issue investigated within the present paper concerns the presence of laminar spots within the turbulent flow. It was shown in the previous contribution [11] that the transverse Taylor to Liepmann scale ratio $\lambda_n / \Lambda$ (see Equation (19)) can be an indicator of external intermittency. We created artificial signals with the spectra prescribed by Equation (20), setting the signal values to a constant over some periods of time (see the upper plots in Figure 27). With this we knew the exact value of the intermittency parameter $\gamma$. Values of $\gamma$ and $\lambda_n / \Lambda$ calculated for the whole signal are presented in Table 1. Even for $\gamma = 0.3$ results compare well. The same estimates calculated for a moving window of size $2T_{ch}$ are presented in the lower plots in Figure 27. Deviations of the mean $\lambda_n / \Lambda$ ratio from unity are accompanied by underprediction of estimates based on the number of zero-crossings (see middle plots in Figure 27.

**Table 1.** Intermittency parameter vs. Taylor-to-Liepmann scale ratio.

| Signal Number | 1 | 2 | 3 | 4 |
|---|---|---|---|---|
| Intermittency parameter | 0.84 | 0.69 | 0.60 | 0.30 |
| $\lambda_n / \Lambda$ | 0.84 | 0.72 | 0.57 | 0.35 |

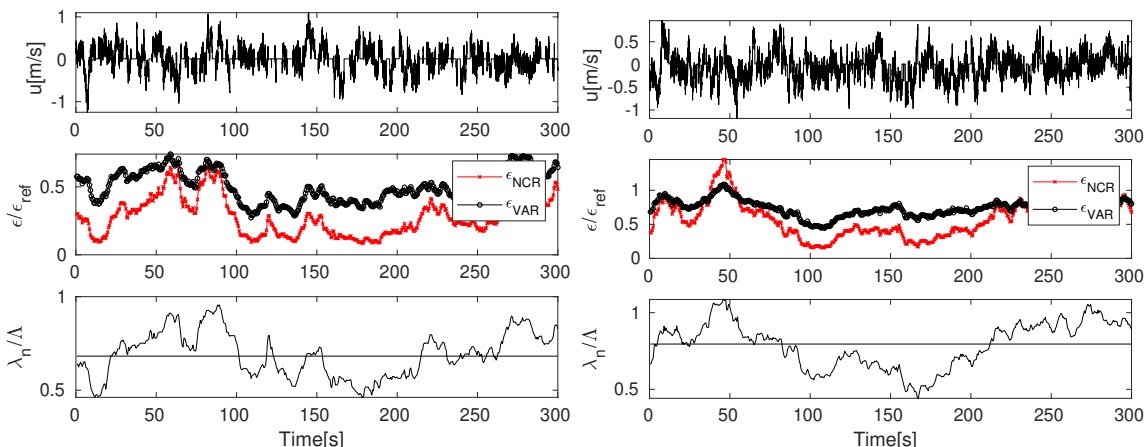

**Figure 27.** Top plots: Part of an artificial signal, middle plots: $\epsilon$ estimates, bottom plots: Taylor-to Liepmann scale ratio. Left column: signal with the intermittency parameter 0.69, right column: signal with the intermittency parameter 0.84.

Analogous study was performed for two sample signals from the POST campaign [15]. Results for a horizontal segment of flight 13 are presented in the left panel in Figure 28. The signal recorded during the flight through the cloud is fully turbulent and the mean $\lambda_n/\Lambda \approx 1$. Right panel of Figure 28 presents analogous results for vertical cloud penetrations during the flight 3. Wind velocity was measured with the frequency $f_s = 10$ Hz. Here, the level of turbulence intensity changes along the flight track and for the considered part of the signal, the calculated mean $\lambda_n/\Lambda \approx 0.8$.

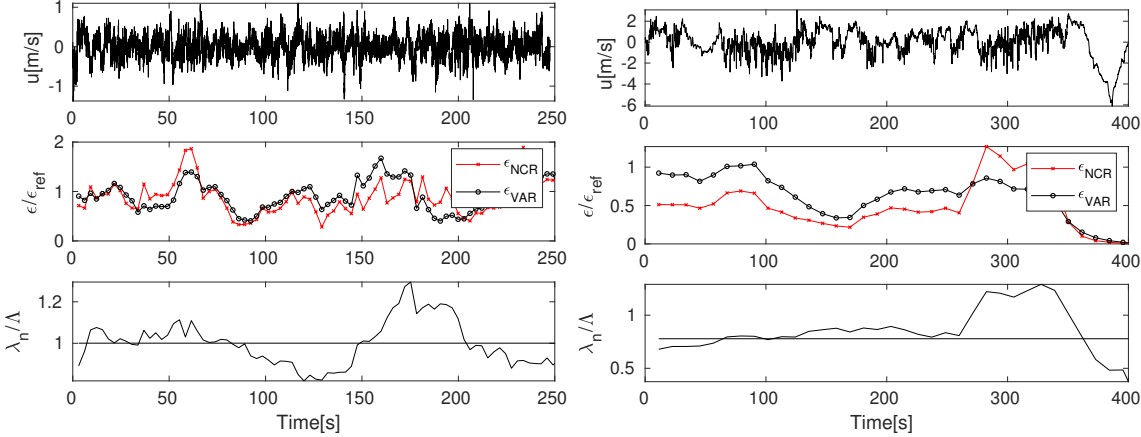

**Figure 28.** Top plots: Part of the POST signals, middle plots: $\epsilon$ estimates, bottom plots: Taylor-to Liepmann scale ratio. Left panel: horizontal segment of the flight 13, right panel: flight 3, signal recorded during vertical cloud penetrations.

## 7. Discussion

Within the present work we investigated performance of different methods of EDR retrieval. We argue, these techniques can be complementary as they respond differently to various types of error. It was observed the power-spectra estimates are sensitive to the aliasing due to finite frequency and the size of the averaging window. This method performed well for the fitting ranges moved towards the small wavenumbers and is expected to give the best estimates in such a case. For the structure function approach, on the other hand, the best results were obtained for fitting ranges moved towards higher wavenumbers (smaller scales). This method appears to be the least sensitive to the finite frequency and the size of averaging window. A clear advantage of the iterative methods proposed in Refs. [10,11] is that, as the Formula (4) is universal, only the value of the effective spectral cut-off $\kappa_{cut} = 2\pi f_{cut}/U$ should be given to estimate $\mathcal{C}_{\mathcal{F}}$ and $\epsilon$ from Equation (10) or Equation (12). These methods are not sensitive to the change of the lower bound of the fitting range. Mean results of $\epsilon_{VAR}$ $\epsilon_{NCF}$ estimates lie often between $\epsilon_{PS}$ and $\epsilon_{SF}$. They are less sensitive to the decrease of the averaging window size and aliasing than $\epsilon_{PS}$, and less sensitive to the change of the fitting range or cut-off frequency than $\epsilon_{SF}$. Results from the number of zero-crossing statistics have usually larger scatter than results of other methods.

In an ideal case results of all estimates should be comparable. It can be concluded from the results of the tests that $\epsilon_{SF}$ tend to be under-predicted in comparison to other methods which can follow from the choice of the fitting range or the aliasing which deteriorates results of $\epsilon_{PS}$ the most and less $\epsilon_{VAR}$ and $\epsilon_{NCF}$. To discuss this issue in more detail we present results of EDR estimates for flight 3 from POST campaign [15], where the data were measured with the low frequency of 10 Hz and atmospheric conditions varied considerably along the flight track. Even though the averaging window $T_{av} = 10T_{ch}$ seemed sufficiently large, results of different methods deviate when the fitting range $f = [0.5 \div 3.5]$ Hz was chosen (see left panel in Figure 29). To minimize the effect of aliasing the fitting range for $\epsilon_{PS}$ and $\epsilon_{NCF}$ was moved to $f = [0.5 \div 2.5]$ Hz and $f_{cut} = 2.5$ Hz for $\epsilon_{VAR}$, whereas the fitting range for the structure function was different and corresponded to $f = [2 \div 5]$ Hz. Such choice should be acceptable, as $\epsilon_{SF}$ was not sensitive to aliasing. As it is seen in the right panel in Figure 29, results compare much

better. Hence, it can be expected that a more reliable value of EDR can be obtained when different methods are compared, as it allows to adjust the bounds of the fitting ranges properly to minimize the bias errors.

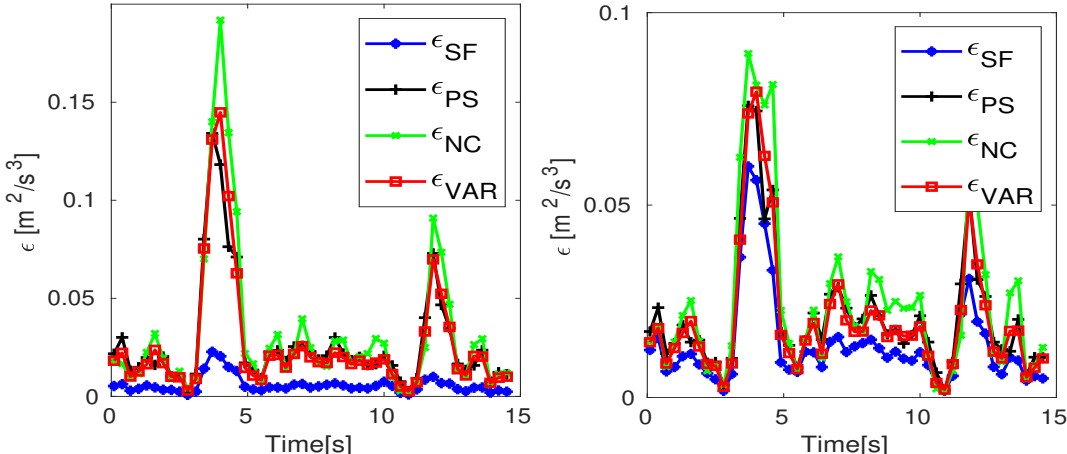

**Figure 29.** EDR estimates for signals measured during POST flight 3 [15] with $f_s = 10$ Hz. **Left panel**: fitting range $f = [0.5 \div 3.5]$ Hz and $f_{cut} = 3.5$ Hz, **right panel**: fitting range corresponding to $f = [2 \div 5]$ Hz for $\epsilon_{SF}$ and $f = [0.5 \div 2.5]$ Hz for $\epsilon_{PS}$ and $\epsilon_{NCF}$, $f_{cut} = 2.5$ Hz for $\epsilon_{VAR}$.

Turbulence in the atmosphere is in reality often far from its idealised, textbook picture. It is non-stationary, anisotropic and affected by buoyancy, local shear and external intermittency, i.e., the presence of laminar patches within turbulent flow. If the fitting range and the averaging window are chosen properly and differences between EDR estimates are still observed, the possible reason are deviations from the Kolmogorov's scaling. It was shown, especially $\epsilon_{SF}$ is in such case under or over-predicted. On the other hand, under-prediction of estimates based on the number of zero-crossings indicates the presence of external intermittency. Hence, deviations between the results can deliver additional informations on the measured physical quantities.

With the developed algorithm for EDR retrieval, it is possible to analyse and compare data from different research campaigns. Our aim is to investigate data from various sources, which will allow to better characterize general characteristic features of a given flow type.

**Author Contributions:** Conceptualization, M.W. and S.P.M.; methodology, M.W.; formal analysis for EDR retrieval techniques, M.W. and A.S.G.; formal analysis and validation for external intermittency, M.W., M.M. and J.N.; writing—original draft preparation, M.W. and S.P.M., writing—review and editing M.W., M.M. and S.P.M. All authors have read and agreed to the published version of the manuscript.

**Funding:** This work received funding from the European Union Horizon 2020 Research and Innovation Programme under the Marie Sklodowska-Curie Actions, Grant Agreement No. 675675. M.W. and S.P.M. acknowledge matching fund from the Polish Ministry of Science and Higher Education No. 341832/PnH/2016.

**Conflicts of Interest:** 'The authors declare no conflict of interest.

## Abbreviations

The following abbreviations are used in this manuscript:

| | |
|---|---|
| EDR | energy dissipation rate |
| PS | power spectra |
| SF | structure function |
| NCF | number of crossings scaling in the inertial range |
| VAR | variance of velocity derivative |
| NCR | number of crossings with spectrum-reconstruction |

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
