# Peer review of "Comparison of Different Techniques to Calculate Properties of Atmospheric Turbulence from Low-Resolution Data"

_atmosphere, doi:10.3390/atmos11020199_

Round 1

Reviewer 1 Report

The manuscripts presents a comparison of four different methods of calculating turbulence characteristics (mainly, the energy dissipation rate} from experimental data - two classic methods and two newer, introduced by some of the authors of the reviewed manuscript. While the paper deals mostly with signal processing techniques and brings little new knowledge to the understanding of atmospheric processes, it can be highly useful to scientists studying experimental data on turbulence. The paper is well organized, clearly written, and contains significant elements of methodological novelty. As I have just a few remarks, mostly of editorial nature, I recommend the paper be published after a minor revision.

I have two remarks of a more general character:
1) It seems to me that all the methods employ at least some exlements of the theory of isotropic turbulence. If I am right, it should be stated in a more explicit way. Otherwise, the differences regarding this assumption should be better exposed.
2) Figs. 4-7 share almost the same caption, with just one element changed. As there are many parameters listed in each of these captions, locating the differences is unnecesarily awkward. I think that a full description should be given only in the first caption whereas the others should be labeled briefly 'As in Fig. 4, but with ....". The same remark applies to subsequent sets of figures (till Fig. 26). Also, the caption would be easier to read if the explanations of the individual elements (curves) was given at the end of the caption and the contents of panels was described prior to it.

English is sufficiently good overall, I have just a few suggestions:

L13: check the use of articles
L17-18: controlled flows ... controlled experiments - i think one of the 'controlled' is extraneous
L38: celebrated Kolmogorov's hypotheses - did he have any 'non-celebrated' ones? (also, L80)
L49-50: "in this analysis, we will analyse"
L62-63: The sentence lacks verb - perhaps it should be combined with the preceding one?
L66: "zero-crossings" - the situation is not completely clear to the reader, I think adding a few explanatory sentences here would help to avoid misunderstandings
L142: "the spectral leakage" or "a spectral leakage"?
L146: "3-4 orders of magnitude worse" -> "smaller by 3-4 orders of magnitude"?
L239: I assume, you mean "sampling frequency"? Or you know the true one?

Author Response

We would like to thank the Referee for his/her comments and suggestions.

We did our best to revise the manuscript, accordingly.

Yes, the underlying assumption for all the methods is the assumption of local isotropy. This allows to estimate EDR from 1D intersections of the three-dimensional velocity field. We addressed this issue in more detail in Section 2, lines 82-87: the sentence starting from "The basis for all the methods is the Kolmogorov's
local isotropy hypothesis ...." etc. We changed the figure captions, as suggested by the Referee We corrected the typos and language mistakes.

Reviewer 2 Report

Referee Report on paper atmosphere-707822 “Comparison of different techniques to calculate properties of atmospheric turbulence from low-resolution data”,  by Waclawczyk et al.

The paper deals with different methods for the assessment of turbulent Energy Dissipation Rate (EDR) from time series. In particular, the authors use four different methods:

power spectra (PS); structure functions (SF); number of zero crossings in the inertial range (NCF); variance of velocity derivative (VAR)

to assess the value of EDR. They initially apply the methods to an “artificial” sample data, obtained through a turbulence model, in order to understand strong and weak points of the methods on data they can manipulate at will, afterwards they move to consider real atmospheric data taken during a campaign of data measurements.

In general, the paper is interesting and the problem the authors deal with is quite important, due to the known limitations in resolution of atmospheric measurements. However, I believe the discussion of the results and the discussion about the main findings of the paper should be improved. In general, whilst the presentation of the problem is rather clear and satisfactory, the results are presented in a rather confusing way and I was myself quite baffled when trying to follow the results for the different methods. I would suggest the authors to:

separate the different plots for the different methods by putting them on different pages; separate the results for the artificial data and the different methods in subsections, and give an in-depth discussion of their findings; enlarge the final part of discussion to stress the conclusions of their work, which is indeed interesting, but given in a rather shortened way. In particular, I would like the authors give a definite statement, a true “recipe”, on which method performs better in which situation and therefore should be used. Or, in case this is not possible, they should give a more extensive discussion about this.

In general, in the plots, it is not always clear whether the plot refers to the artificial data or to the real data. This is a fundamental information that should be made clear.

Along with the previous observations, which I intend as a way to improve the general readability of the paper, there are some “technical” points on which I would like to hear some discussion from the authors.

First of all, at the end of page 8, lines 167-173, the authors discuss briefly the sample of real data they use in their analysis. However, they estimate the average velocity of the flow as U ~ 55 ms^-1 and the typical amplitude of turbulent fluctuations as u’ ~ 0.35 ms^-1. Now, with a level of fluctuations around 0.4%, this flow does not appear to be much turbulent. Is there anything I am missing? In the top panel of fig. 3, the authors show “part of the signal”, but of which “signal”? The artificial data produced by the turbulence model or the real data? I believe the former one. It would be interesting also to put a plot with the real data, at least to compare how the real data and artificial data compare each to the other. Same situation is in Fig. 2, where a power spectrum of the signal is shown. But from which data? I understood that is the signal from the artificial data (but I may be wrong!), but that should be explicitly stated somewhere. Putting also a spectrum of the real data would be also convenient, since that would clarify at once whether the dataset of real measurements is really enough turbulent or not.

Secondly, the authors discuss the presence of external intermittency in atmospheric turbulence. Now, several papers (e.g. Capet et al., Journ. Phys. Oceanogr. (2008), D’Asaro et al., Journ. Phys. Oceanogr. (2006) for data analysis in the oceans and, more recent, Feraco et al., EPL (2018), for a numerical example in stratified atmospheric flows) found that external intermittency is especially present at large scales in the vertical component of the velocity field, while the horizontal components have rather Gaussian (or even subgaussian) velocity statistics. As far as I understood, the authors just speculate about the suitability of their analysis to catch the presence of external intermittency in their artificial data set but they say nothing concerning the actual presence of external intermittency in real data. This would be a very interesting point to discuss in the paper. At least, in my opinion, the authors should briefly discuss why they limit themselves to artificial data without reference to real data. Is there any special reason for this?

Finally several typos are present in the manuscript (see, for instance, pag. 2, line 35, “mili- or centimeter” should be “milli- or centimeter”; line 52, “([7,8]” should be “([7,8])”, and many more) and some notation look a little bit confusing in some cases. For instance, in eq.s (7-8), wouldn’t be better to indicate the longitudinal and transverse structure functions with $D_\parallel$ and $D_\perp$ instead of $D_l$ and $D_n$, since $n$ is used in lines 97-98 to indicate another quantity? Or, when indicating a range (see, for instance, $f=[10 20]Hz$ at line 193, pag. 9), I believe a notation like: “f=[10 \div 20] Hz$ would be much better. In general, I would suggest the authors to thoroughly re-read the manuscript to improve a little bit both the clarity of the text and the notation.

When all these points have been clarified, in my opinion the article is well suitable to be published in Atmosphere.

Author Response

We would like to thank the Referee for his/her comments and suggestions. We did our best to improve the manuscript accordingly.

We changed the order of figures, as suggested. We start from results for the power-spectra and describe them in detail in Subsection 4.2.1, next, results for structure functions are presented in Subsection 4.2.2. This is followed by Subsection 4.2.3 with results of number-of crossings method and finally, results of iterative method are discussed in Subsection 4.2.4. We enlarged the final discussion and added a new figure 28, which illustrates advantage of using different EDR estimates in parallel. This helps to find optimal (or correct) settings, like e.g. the fitting ranges. We added information on plots to specify which of them refer to artificial and which to real data. In particular, Figs. 1 and 2 are plotted for the POST data. The true air speed U, we refer to on page 8 is the magnitude of the vector difference between the aircraft velocity and the wind velocity (we added this information in the revised manuscript). The mean wind velocity, on the other hand is about 4m/s. With this, the level of fluctuations is about 10%.   In Fig. 3 the artificial data are presented. We put the real data later in the manuscript, in Fig. 27. We added more discussion on intermittency in Section 6. We added references suggested by the Referee and additional references 24,29,30. We calculated the Taylor-to-Liepmann scale ratio for exemplary POST signals. Results are presented in Fig. 29. The signal investigated in Section 5 was measured during a horizontal flight through a cloud. Hence, its intermittency ratio is close to unity. We expect the intermittency could be detected during vertical penetrations of the cloud. Sample results of such signal are presented on the right panel in Fig. 29. However, a detailed investigation of this issue is left for further work.
We introduced the minor corrections as suggested by the Referee.

Round 2

Reviewer 2 Report

Referee Report on paper atmosphere-707822 “Comparison of different techniques to calculate properties of atmospheric turbulence from low-resolution data”,  by Waclawczyk et al.

I believe the authors did a very good job in this revised version of the paper which is now much clearer, in my opinion, and more readable. All my doubts have been clarified and I believe that now the paper is well suitable for publication in "Atmosphere".